# Audiovisual AR concepts for laparoscopic subsurface structure navigation

Fabian Joeres*
Otto-von-Guericke University

David Black†
Fraunhofer Institute for
Digital Medicine MEVIS
Otto-von-Guericke University

Seyedsina Razavizadeh ‡
Otto-von-Guericke University

Christian Hansen§
Otto-von-Guericke University

## ABSTRACT

The identification of subsurface structures during resection wound repair is a challenge during minimally invasive partial nephrectomy. Specifically, major blood vessels and branches of the urinary collecting system need to be localized under time pressure as target or risk structures during suture placement. This work presents concepts for AR visualization and auditory guidance based on tool position that support this task. We evaluated the concepts in a laboratory user study with a simplified, simulated task: The localization of subsurface target points in a healthy kidney phantom. We evaluated the task time, localization accuracy, and perceived workload for our concepts and a control condition without navigation support. The AR visualization improved the accuracy and perceived workload over the control condition. We observed similar, non-significant trends for the auditory display. Further, clinically realistic evaluation is pending. Our initial results indicate the potential benefits of our concepts in supporting laparoscopic resection wound repair.

**Keywords:** Augmented reality; audio navigation; laparoscopic surgery; partial nephrectomy; visualization.

**Index Terms:** Human-centered computing—Visualization—; Human-centered computing—Human computer interaction (HCI)—Interaction paradigms—Mixed / augmented reality; Human-centered computing—Human computer interaction (HCI)—Interaction devices—Sound-based input / output; Applied computing—Life and medical sciences——

## 1 INTRODUCTION

### 1.1 Motivation

The field of augmented reality (AR) for laparoscopic surgery has inspired broad research over the past decade [2]. This research aims to alleviate the challenges that are associated with the indirect access in such operations. One challenging operation that has attracted much attention from the research community is laparoscopic or robot-assisted partial nephrectomy (LPN/RPN) [17,20]. LPN/RPN is the standard treatment for early-stage renal cancer. The operation's objective is to remove the intact (i.e., entire) tumor from the kidney while preserving as much healthy kidney tissue as possible. Three challenging phases in this operation can particularly benefit from image guidance or AR navigation support [20]: i) the management of renal blood vessels before the tumor resection, ii) the intraoperative resection planning and the resection, iii) the repair of the resection wound after the tumor removal. Management of renal blood vessels includes the decision which arteries are to be

---

*e-mail: fabian.joeres@ovgu.de
†e-mail: david.black@mevis.fraunhofer.de
‡e-mail: seyedsina.razavizadeh@ovgu.de
§e-mail: christian.hansen@ovgu.de

clamped for the resection and the localisation, dissection, and clamping of those arteries. The resection wound repair phase consists of two main steps: The surgeon has to identify major lesions of blood vessels and in the urinary duct system and set individual sutures to close these. In the second step, an overall suture is placed to close the wound. The surgeon runs this suture closely under the wound's surface and needs to avoid damaging subsurface blood and urinary vessels in this area.

Although numerous solutions have been proposed to support urologists during the first two phases [20], no dedicated AR solutions exist for the third. Specifically, urologists need to identify major blood vessels or branches of the urinary collecting system that have been severed or that lie closely under the resection wound's surface and could be damaged during suturing. One additional challenging factor is that this surgical phase is performed under time pressure. This is due to the risk of ischemic damage (i.e. damage from a lack of blood perfusion) if relevant arterial branches have been clamped or to increased blood loss if they have not. There are some technical challenges in providing correct AR registration and meaningful navigation support during this phase. One challenge that affects the visualization of AR information is the removal of renal tissue volume that leaves an undefined tissue surface that is inside the original organ borders. In this work, we present an AR visualization and an auditory display concept that rely on the position of a tracked surgical tool to support the urologist in identifying and locating subsurface structures. We also report a preliminary proof-of-concept evaluation through a user study with an abstracted task. AR registration and the clinical evaluation of our concepts lie outside the scope of this work.

### 1.2 Related work

Multiple reviews provide a comprehensive overview of navigation support approaches for LPN/RPN [3,17,20]. Although no dedicated solutions exist to support urologists during the resection wound repair phase, one application has been reported, in which the general AR model of intrarenal structures was used during renorrhaphy [25]. This approach, however, does not address the unknown resection wound surface geometry and potential occlusion issues. Moreover, multiple solutions have been proposed to visualize intrarenal vascular structures. These include solutions in which a preoperative model of the vascular structure is rendered in an AR overlay [24,31]. This may be less informative after an unknown tissue volume has been resected. Other methods rely on real-time detection of subsurface vessels [1,18,30]. However, these are unlikely to perform well when the vessels are clamped (suppressing blood flow and pulsation) or when the organ surface is occluded by blood. Outside of LPN/RPN, such as in angiography exploration, visualization methods have been developed to communicate the spatial arrangement of vessels. These include the chromadepth [29] and pseudo-chromadepth methods [21,27], which map vessel depth information to color hue gradients. Kersten-Oertel et al. [22] showed that color hue mapping, along with contrast grading, performs well in conveying depth information for vascular structures. The visualization of structures based on tool position has inspired work both inside and outside of the field of LPN/RPN: Singla et al. [28] proposed visualizing the tool position

in relation to the tumor prior to resection in LPN/RPN. Multiple visualizations have been proposed for the spatial relationship between surgical needles and the surrounding vasculature [15]. However, these visualizations explore the application of minimally-invasive needle interventions where the instrument is moving in between the structures of interest.

In addition to visual approaches to supporting LPN/RPN as well as other navigated applications, recent works have shown that using sound to augment or replace visual cues can be employed to aid task completion. By using so-called auditory display, changes in a set of navigation parameters can be mapped to changes in parameters of a real-time sound synthesizer. This can be found in common automobile parking assistance systems: the distance of the automobile to a surrounding object is mapped to the inter-onset-interval (i.e., the time between tones) of a simple synthesizer. Using auditory display has been motivated by the desire to increase clinician awareness, replacing the lost sense of touch when using teleoperated devices, or help clinicians correctly interpret and follow navigation paths. There have been, however, relatively few applications of auditory display in medical navigation. Evaluations have been performed for radiofrequency ablation [5], temporal bone drilling [8], skull base surgery [9], soft tissue resection [13], and telerobotic surgery [6, 23]. These have shown auditory display to improve recognition of structure distance and accuracy and diminish cognitive workload and rates of clinical complication. Disadvantages have included increased non-target tissue removal and more lengthy task completion times. For a thorough overview of auditory display in medical interventions, see [4].

## 2 NAVIGATION METHODS

We pursued two routes to provide navigation content to the urologist: The first approach is the AR visualization of preoperative anatomical information in a video-see through setting. The second approach is an auditory display.

### 2.1 AR visualization

Our AR concept aims to provide information about intrarenal risk structures to the urologists. We, therefore, based our visualization on preoperative three-dimensional (3D) image data of the intrarenal vasculature and collecting system. These were segmented and exported as surface models. We assumed that the resection volume and resulting wound geometry are unknown. Simply overlaying the preoperative models onto the laparoscopic video stream would include all risk structures that were resected with the resection volume. We, therefore, propose a tool-based visualization. In this concept, only information about risk structures in front of a pointing tool are rendered and overlaid onto the video stream. To this end, the urologist can place a spatially tracked pointing tool on the newly created organ surface (i.e., resection ground) and see the risk structures beneath. We placed a virtual circular plane perpendicular to the tool axis with a diameter of 20mm around the tooltip. The structures in front of this plane (following the tool direction) are projected orthogonally onto the plane and rendered accordingly. The two different structure types are visualized with two different color scales (Figure 1a). The scales visualize the distance between a given structure and the plane. The scale ends are equivalent to a minimum and maximum probing depth that can be set for different applications. The scale hues were selected based on two criteria: Firstly, we investigated which hues provide good contrast visibility in front of laparoscopic videos. Secondly, the choice of yellow for urinary tracts and blue-magenta for blood vessels is consistent with conventions in anatomical illustrations and should be intuitive for medical professionals. For the urinary tract, color brightness and transparency are changed across the spectrum. For the blood vessels, color hue, brightness, and transparency are used. These color spectrums aim to combine the color gradient and fog concepts

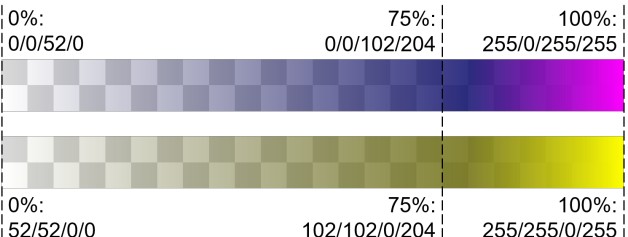

(a) color spectrum for blood vessels (top) and urinary tract (bottom). The color values are in RGBA format.

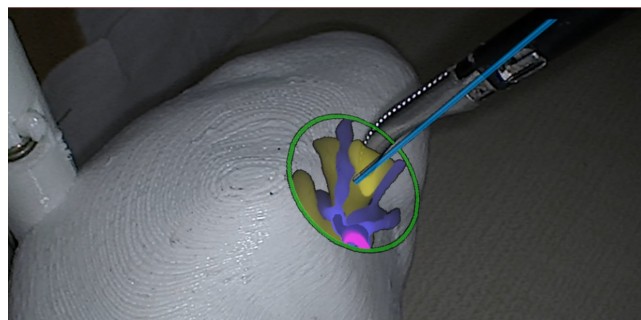

(b) Laparoscopic view of a printed kidney phantom with the visual AR overlay.

Figure 1: AR visualization.

that were identified as promising approaches by Kersten-Oertel et al. [22]. An example for the resulting visualization (using a printed kidney phantom) is provided in Figure 1b. The blue line marks the measured tool axis.

### 2.2 Audio navigation

After iterative preliminary designs were evaluated informally with 12 participants, an auditory display consisting of two contrasting sounds was developed to represent the structures. The sound of running water was selected to represent the collecting system, and a synthesized tone was created to represent the vessels. The size and number of the vessels in the scanning area are encoded in a three-level *density* score. Density is then mapped to the water pressure for the collecting system, and the tone's pitch for vessels, with higher pressure and pitch indicating a denser structure. Finally, the rhythm of each tone is a translation of the distance between the instrument tip and the closest point on the targeted structure, with a faster rhythm representing lesser distance. To express the density of the collecting system, the water pressure is manipulated to produce three conditions, i.e., low, medium, and high pressure; representing low, medium, and high density. The water tone is triggered every 250, 500, and 2000ms, depending on the distance: inside, close, and far. A distant structure resembles an uninterrupted flow of water, and a nearby structure is heard as rhythmic splashes. Inside the structure, a rapid splashing rhythm is accompanied by an alert sound.

### 2.3 Prototype implementation

We implemented our overall software prototype and its visualization in Unity 2018 (Unity Software, USA). The auditory display was implemented using Pure Data [26].

#### 2.3.1 Augmented reality infrastructure

The laparoscopic video stream was generated with an EinsteinVision© 3.0 laparoscope (B. Braun Melsungen AG, Germany) with a 30° optic in monoscopic mode. We used

standard laparoscopic graspers as a pointing tool. The camera head and the tool were tracked with a NDI Polaris Spectra passive infrared tracking camera (Northern Digital Inc., Canada). We calibrated the laparoscopic camera based on a pinhole model [32] as implemented in the OpenCV library[1] [7]. We used a pattern of ChArUCo markers [12] for the camera calibration. The external camera parameters (i.e., the spatial transformation between the laparoscope's tracking markers and the camera position) were determined with a spatially tracked calibration body. The spatial transformation between the tool's tracking markers and its tip was determined with pivot calibration using the NDI Toolbox software (Northern Digital Inc.). The rotational transformation between the tracking markers and the tool axis was measured with our calibration body. The resulting laparoscopic video stream with or without AR overlay was displayed on a 24 inch screen. AR registration for this surgical phase was outside of scope for this study. The kidney registration was based on the predefined spatial transformation between our kidney phantom and its tracking geometry (see *Study setup*).

### 2.3.2 AR visualization implementation

The circular plane was placed at the tooltip and perpendicular to the tool's axis as provided by the real-time tracking data. The registration between the visualization and the camera were provided by the abovementioned tool and camera calibration and the real-time tracking data. The plane was then overlaid with a mesh with a rectangular vertex arrangement. The vertices had a density of 64 pts/mm$^2$ and served as virtual pixels. We conducted a ray-casting request for each vertex. For each ray that hit the surface mesh of the structures in our virtual model, the respective vertex was colored according to the type and ray collision distance of that structure. The visualization was permanently activated in our study prototype.

### 2.3.3 Auditory display implementation

The synthesized tone contrasts the water sound to ensure distinction between the sounds. The synthesized sound is created from the base frequencies of 65.4 Hz, 130.8 Hz, and 261.6 Hz (C2, C3, and C4 notes) and harmonized by each frequency's first to eighth harmonics, creating a complex tone. The density of the vessels is measured on ray casting requests that are equivalent to the visual implementation. The number of virtual pixels that would depict a given structure type determin the density for that type. This density is then encoded in the pitch of the tone, meaning that 65.4 Hz, 130.8 Hz, and 261.6 Hz (C2, C3, and C4 notes) represent low, medium, and high density, respectively. The repetition time of the tones expresses the distance between the instrument tip and the closest point on the targeted vessel. Similar to the water sound, a continuous tone represents a far-away vessel, while a close vessel is heard as the tone being repeated every 500 ms with a duration of 400ms. Being inside the vessel triggers an alert sound played every 125 ms accompanied by the tone every 250 ms.

## 3 EVALUATION METHODS

We conducted a simulated-use proof-of-concept evaluation study with N=11 participants to investigate whether our concepts effectively support the urologists in locating subsurface structures in laparoscopic surgery.

### 3.1 Study task

The specific challenges of identifying relevant subsurface structures for suture placement in resection wound repair are difficult to replicate in a laboratory setting. We devised a study task that aimed to imitate the identification of specific structures beneath an organ

---

[1]We used the commercially available *OpenCV for Unity* package (Enox Software, Japan)

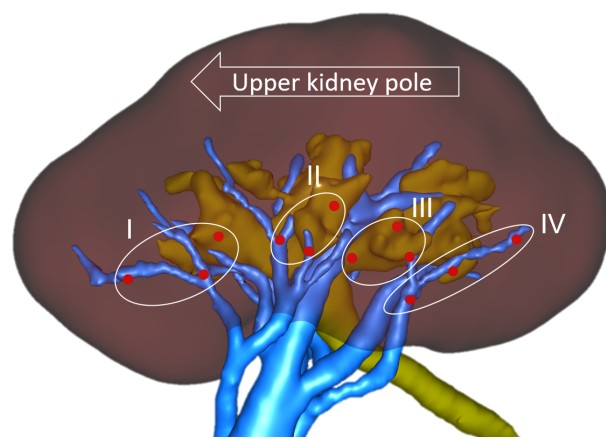

Figure 2: Virtual kidney model with the target point clusters. The model is shown from a medial-anterior perspective, corresponding to the participant's position.

surface: Participants were presented with a printed kidney phantom in a simulated laparoscopic environment. We also displayed a 3D model of the same kidney on a 24 inch screen. This virtual model included surface meshes of the vessel tree and collecting system inside that kidney (Figure 2). Participants could manipulate the view of that model by panning, rotating, and zooming. For each study trial, we marked a point on the internal structures (a blood or urine vessel) in the virtual model with a red dot (Figure 2). The target points were arranged into four clusters to prevent familiarization with the target structures throughout the experiment. The participants were then asked to point the surgical tool at the location of that subsurface point in the physical phantom as accurately and as quickly as possible by placing the tool on the surface and orienting it such that the tool's direction pointed towards the internal target point.

### 3.2 Study design

Our study investigated the impact of the visual and auditory support on the performance and perceived workload of the navigation task. We examined two independent variables with two levels each ($2 \times 2$ design): The presence or absence of the visual support and the presence or absence of the auditory support. The condition in which neither support modality was present was the control condition. Three dependent variables were measured and analyzed: Firstly, we measured the task completion time. Time started counting when the target point was displayed. It stopped when participants gave a verbal cue that they were confident they were pointing at the target as accurately as possible. Secondly, we measured how accurately they pointed the tool. Accuracy was measured as the closest distance between the tool's axis and the target point (point-to-ray distance). Finally, we used the NASA Task Load Index (NASA-TLX) [14] questionnaire as an indicator for the perceived workload. The NASA-TLX questionnaire is based on six contributing dimensions of subjectively perceived workload. The weighted ratings for each dimension are combined into an overall workload score.

### 3.3 Study sample

Eleven (11) participants took part in our study (six females, five males). All participants were medical students between their third and fifth year of training. Participants were aged between 24 and 33 years (median = 25 years). All participants were right-handed. Four participants reported between one and five hours of experience with laparoscopic interaction (median = 3h) and seven participants reported between one and 15 hours of AR experience (median = 2h).

Finally, eight participants reported to be trained in playing a musical instrument. No participants reported any untreated vision or hearing impairments.

### 3.4 Study setup

The virtual kidney model and its physical phantom were created from a public database of abdominal computed tomography imaging data [16]. We segmented a healthy left kidney using 3D Slicer [11] and exported the parenchymal surface, the vessel tree, and the urinary collecting system as separate surface models. The parenchymal surface model was printed with the fused deposition modeling method and equipped with an adapter for passive tracking markers (Figure 3a). The phantom was placed in a cardboard box to simulate a laparoscopic working environment (Figure 3b). The screen with the laparoscopic video stream was placed opposite the participant and the screen with the virtual model viewer was placed to the participant's right. A mouse was provided to interact with the model viewer and a standard commercial multimedia speaker was included for the auditory display. The overall study setup is shown in Figure 4.

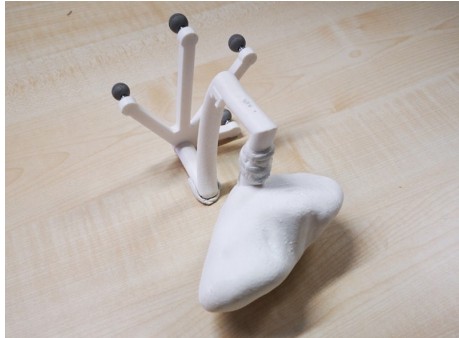

(a) Kidney phantom with tracking marker adapter.

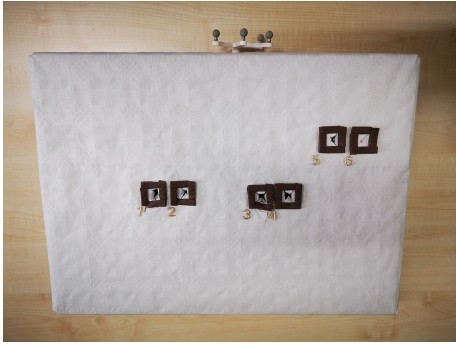

(b) Cardboard box with tool holes.

Figure 3: Components of the simulated laparoscopic environment.

### 3.5 Study procedure

Participants' written consent and demographic data were collected upon arrival. The participants then received an introduction to the visualization and auditory display of the data. Participants conducted one trial block per navigation method. In each trial block, they were asked to locate the three points of one cluster, with one trial per point. After each trial block, one NASA TLX questionnaire was completed for the respective navigation method. The order of the navigation methods and the assignment between the point clusters and the navigation methods were counterbalanced. The order in which the points had to be located within each trial block was permutated.

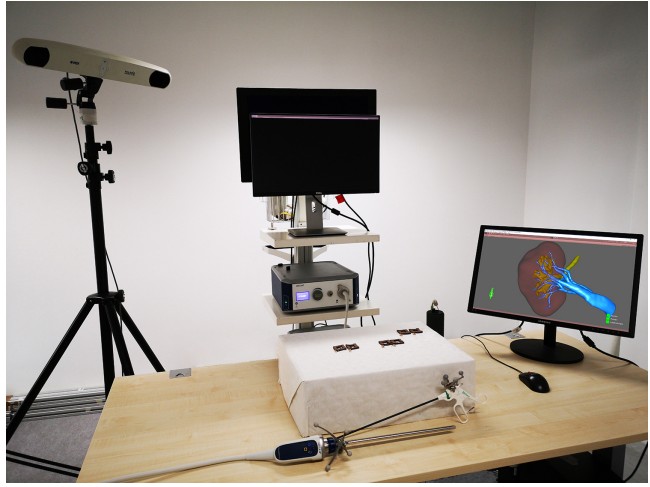

Figure 4: Overall study setup.

### 3.6 Data analysis

During initial data exploration, we noticed a trend that participants took more time to complete the task in the first trial they attempted with each method than in the second and third trials. Therefore, the first trial for each method and participant was regarded as a training trial and excluded from the analysis. The data (time and accuracy) from the remaining two trials from each block were averaged and a repeated-measures two-way analysis of variance (ANOVA) was conducted for each dependent variable.

### 4 RESULTS

The descriptive results for the three dependent variables are listed in Table 1. We found significant main effects for the presence of visual display onto the accuracy ($p < 0.001$) and the NASA TLX rating ($p = 0.03$). The ANOVA results are listed in Table 2. The significant effects are plotted in Figure 5.

The most evident result from our evaluation is that the visual display increases the accuracy and reduces the perceived workload of identifying subsurface vascular and urinary structures in our simplified task. At the same time, the visual display method did not reduce the task completion time. Generally, there were non-significant trends that all tested conditions with visual or auditory display performed more accurately and tended to cause a lower perceived workload. However, the navigation support conditions tended to perform less quickly than the control condition. This may be due to the fact that the required mental spatial transformations are reduced, but a greater amount of information needs to be processed by the participants.

This explanation is also supported by the result that the combined auditory and visual display performed worse than the visual-only condition within our sample. While this trend is not statistically significant, it poses a question: Were the auditory display designs somewhat misleading or distracting, or is the combination of multimodal channels for the *same* information in itself potentially hindering in this task? The trend of auditory support performing slightly better than the control condition within our sample (no significance) may indicate that the latter explanation is more likely. Another aspect may be users' lower familiarity with auditory navigation than visual cues. Further training and greater participant experience may also reduce the difference in performance between he visual and auditory navigation aids.

The absolute values we measured for our dependent variables are less meaningful than the comparative effects we found for our navigation conditions. Multiple design factors limit the clinical validity

Table 1: Descriptive results for all dependent variables. All entries are in the format <mean value (standard deviation)>.

| Navigation condition | Task completion time [s] | Accuracy [mm] | NASA-TLX |
|---|---|---|---|
| No support | 29.92 (18.59) | 12.54 (4.21) | 14.14 (2.18) |
| Auditory support | 41.44 (22.9) | 9.69 (5.14) | 12.93 (3.46) |
| Visual support | 36.02 (19.21) | 4.39 (3.49) | 10.87 (3.86) |
| Auditory and visual support | 38.12 (22.48) | 6.45 (5.08) | 11.93 (3.3) |

Table 2: ANOVA results for all variables. AD: Auditory display, VD: Visual display. All cells are in the format <F value (degrees of freedom); p value>.

| Dependent variable | Main effect AD | Main effect VD | Interaction AD:VD |
|---|---|---|---|
| Task completion time | 1.41(1,10); 0.263 | 0.17(1,10); 0.688 | 1.47(1,10); 0.253 |
| Accuracy | 0.11(1,10); 0.748 | **28.01(1,10); <0.001\*** | 2.67(1,10); 0.133 |
| NASA TLX | 0.01(1,10); 0.911 | **6.35(1,10); 0.03\*** | 1.47(1,10); 0.253 |

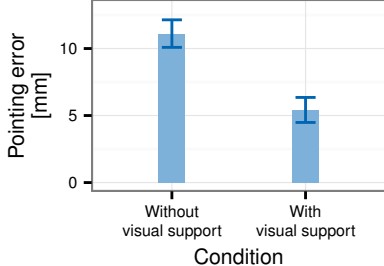

(a) Visual display main effect on the pointing accuracy.

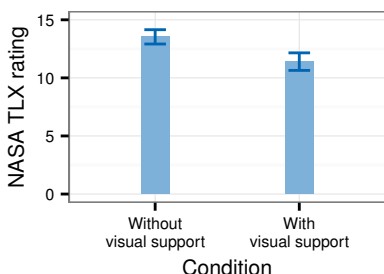

(b) Visual display main effect on the NASA TLX rating.

Figure 5: Significant ANOVA main effects. The error bars represent standard errors.

of our study, including the exclusion of a registration pipeline. This means that the absolute task time or pointing error may well deviate from the reported descriptive results.

## 5 DISCUSSION

Our evaluation yielded preliminary and successful proof-of-concept results for the audiovisual AR support for resection wound repair. The results indicate that audio guidance may be helpful but the benefit could not be significantly within our sample. However, there are limitations to the clinical validity of our prototypes and our study setup.

First and foremost, the study task is an abstraction of the actual surgical task: The surgical task requires not only the identification of major subsurface structures but also the judgment and selection of a suture path. This task limitation went along with an abstract laparoscopic environment and surgical site. Our kidney phantom imitated an in-vivo kidney only in its geometric properties. The color, biomechanical behavior, and surgical surroundings did not resemble their real clinical equivalents. Moreover, the phantom was simplified in that it was based on an intact kidney rather than containing a resection wound. While this simplification is an additional limitation to our study's clinical validity, we believe that introducing a phantom with a resection bed will only be meaningful in combination with a more complex simulated task. This is because, in a realistic setting, the urologist will be familiar with the wound and aware of potential landmarks (like intentionally severed vessels) to help navigation. This would not have been the case in our simplified task and for our participants. One further step in improving the phantom for increased realism may be the simulation of the deformation that occurs. This may be achieved by producing one preoperative phantom and one intraoperative phantom based on simulated intraoperative deformation (e.g., using the Simulation Open Framework Architecture [10])

Another aspect to improve the clinical validity of our evaluation could be a more realistic task. The most valid performance parameter, however, would be the frequency of suture setting errors. Because these are not very frequent, the study would require a large sample consisting of experienced urologists. This is logistically challenging. We, therefore, regard our preliminary evaluation as a good first indication for the aptitude of our navigation support methods.

Future evaluation with a more realistic phantom and task should include the overlay of AR structures on the simulated resection wound as an (additional) reference condition. Moreover, AR registration was excluded from our study's scope to focus the investigation on the tested information presentation methods. A dedicated registration method for post-resection AR has been previously proposed [19]. This could be combined with the dedicated AR concepts reported in this article for future, high-fidelity evaluations. The tested AR visualization was well suited for the abstracted task in our proof-of-concept evaluation. In the clinical context, a semi-transparent display of our visualization may be better suited to prevent occlusion of the relevant surgical area. This occlusion can further be reduced by providing a means to interactively activate or deactivate the visualization.

The participants were medical students with limited laparoscopic experience: They were less trained in the spatial cognitive processes that are involved in laparoscopic navigation than the experienced urologists who would be the intended users for a support system like ours. Thus, the navigation methods presented in this article will need to be further evaluated in clinically realistic settings. This may include testing on an in-vivo or ex-vivo human or porcine kidney phantom. This, however, requires an effective AR registration that

is compromised by the time pressure (for in-vivo phantoms) or post-mortem deformation in ex-vivo phantoms.

Beside more clinically valid evaluation, some other research questions arise from our work: Firstly, some design iteration and comparison should be implemented to evaluate whether the limited success of our auditory display was due to the specific designs or due to a limited aptitude of the auditory modality for such information. Secondly, further visualizations should be developed and compared with our first proposal to identify an ideal information visualization. Finally, it should be investigated whether other procedures with soft tissue resection (e.g., liver or brain surgery) may benefit from similar navigation support systems for the resection wound repair.

## 6 CONCLUSION

This work introduces and tests an audiovisual AR concept to support urologists during the resection wound repair phase in LPN/RPN. To our knowledge, these are the first dedicated solutions that have been proposed for this particular challenge. These concepts have been preliminarily evaluated in a laboratory-based study with an abstracted task. Although the results only represent a proof-of-concept evaluation, we believe that they indicate the potential of our concepts. The next steps for this work include the integration of a targeted AR registration solution and the integrated prototype's evaluation in a clinically realistic setting. Pending this work, we believe that the concepts presented in this article sketch a promising path to a clinically meaningful AR navigation system for minimally invasive, oncological resection wound repair.

### ACKNOWLEDGMENTS

This work has been funded by the EU and the federal state of Saxony-Anhalt, Germany, under grant number ZS/2016/10/81684.

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
