# OpenReview forum: "Audiovisual AR concepts for laparoscopic subsurface structure navigation"
_graphicsinterface.org/Graphics_Interface/2021/Conference/Second_Cycle — GI 2021_

### Official Review · Reviewer_TjQA · 2021-04-17
**Would audiovisual AR help in the OR? This paper leaves many unanswered questions**

**Rating:** 7
**Confidence:** 2

**Review:**

The paper presents a user study evaluating AR-augmented techniques for laparoscopic wound repair of a kidney after tumor removal. Specifically, the surgical procedure is called laparoscopic or robot-assisted partial nephrectomy, abbreviated “LPN/RPN”.

Although augmented reality is not within my area of expertise, I was excited to read this paper. I had hoped to learn about the tracking and imaging techniques used. However, such AR registration concepts are not within this paper’s scope. Instead, this paper serves as a proof of concept and evaluation of the AR techniques. The paper presents this limitation in the Introduction, but I would prefer it also mentioned briefly in the Abstract.

Despite not providing a solution to the AR registration challenge, the paper presents many techniques for similar procedures, their pros and cons, and how they might apply to LPN/RPN. The related works section seems concise, yet detailed and thorough. Additionally, the general discussion section and conclusion suggest that the authors already have a registration solution paper under review and a clinical evaluation planned. This paper’s focus only on the evaluation of AR techniques in a simulated setting seems to be a design choice, rather than an omission of key implementation details and validation.

The navigation method details and prototype implementation description seem sufficient for replicating the study. If other, more knowledgeable reviewers find some information lacking, I am confident that the authors can add the it in a minor revision.

In the study sample section, I suggest adding a sentence or two justifying the reporting of musical instrument experience. I know that there are studies correlating the manual dexterity of surgeons and musicians. I believe there are similar studies on surgeons who play video games. Asking about video game experience (or other manual dexterity tasks) might be beneficial in future studies.

When there are more than 2 experiment conditions, a significant ANOVA result indicates that there a significant difference amongst the conditions, but not which specific conditions exhibit the significant result. In such a situation, analysis should include posthoc tests to identify the one or more pairwise comparisons that were significant. Additionally, I recommend performing ANOVA on the between-subject factor “order” (i.e., the order of navigation methods). A non-significant result would confirm that counterbalancing effectively prevented asymmetric skill transfer.

There are a few recommended typographic edits:
-	No need to provide bold headings in the Abstract
-	There are some non-wrapping lines in references 13 and 15
-	Is there a missing word, Section 5.2, second sentence?

This paper represents an incomplete but important step in an AR system for laparoscopic renorrhaphy. It evaluates interaction techniques for their potential benefits and drawbacks, and justifies further research and complex clinical trials. The contribution of this paper is consistent with its length, and it is excellently written. I am hereby recommending this paper for publication.

---

### Official Review · Reviewer_cMYJ · 2021-05-03
**Nice idea, appropriate for HCI?**

**Rating:** 6
**Confidence:** 3

**Review:**

SUMMARY AND CONTRIBUTION

This paper contributes the design of an augmented reality (AR) based visualization and a related auditory display concept to aid surgeries related to nephrectomy (kidney removal) -- specifically for navigating laparoscopic subsurface structures during resection wound repair. The target user group is urologists who can use this guidance during a minimally invasive partial nephrectomy. A user study with 11 medical students assessed the effectiveness of both the AR visualization and the audio-based navigation methods using virtual kidney models. The key findings showed that both the AR visualization and audio navigation improved the accuracy and perceived workload compared to the baseline, but that the results were only significant for the AR visualization method.

ORIGINALITY AND SIGNIFICANCE OF THIS WORK

The overall concept of using AR and audio navigation techniques to support the way urologists can identify and locate subsurface structures during the wound repair phase of kidney removal seems to be useful (judging as a non-medical expert). The methodological approach of proposing two different design ideas for navigation and comparing them against a baseline seems to be appropriate. I can appreciate the challenges of working with domain experts in medicine, particularly specialists, so I do appreciate the authors’ efforts in recruiting medical students as proxies. It can be also be challenging to come up with appropriate tasks that can simulate the intended behaviour in the system and overall I thought that the authors did really well with their prototype and simulation in the study.

My problem with the paper the way it is currently written is that it is written for a very niche audience who can understand the significance (and related terminologies) of challenges in nephrology. But, I am not able to clearly articulate what the HCI innovation is in the audio/visual approach that improves the state-of-the-art or solves a new problem? There hasn’t been much effort to contextualize the contribution of this work in the broader field of HCI or Info Viz.


QUALITY AND VALIDITY

The description of the system and the interaction techniques are lacking details to get a clear picture of how this actually works (the video was more clear, but the writing does not convey the same clarity). An HCI audience would find it more useful to know how the design space was explored, how were different considerations made, do these designs generalize beyond tasks in nephrology, etc. In the absence of a formative study, it is difficult to know why this was the right approach to tackling this design problem.

The study with 11 participants doesn’t seem to be very through and the empirical results also seem a bit inconclusive in terms of the overall effectiveness of the audiovisual approach (other than the AR part helping improve accuracy/ perceived workload). However, I do realize that it was still at the stage of proof-of-concept and I do appreciate that the authors at least tried to do a realistic evaluation. It would have been helpful to see some qualitative data from the participants about their perceptions and where their struggles were in interacting with the techniques.


CLARITY

Overall, the paper is a bit difficult to read as it is laden with medical and technical terminologies. It would be helpful to make the new design contributions more clear and tease out the key takeaways for an HCI audience.

I also found that the results and discussion sections were a bit muddled up (where the discussion is actually reporting the key results while the result section only points at figures). I would suggest presenting the key findings in the results and use the discussion for interpreting the results and the overall takeaways of this work.


Overall, I do think that the GI audience can find this type of paper to be interesting and valuable. But, with the way the paper is currently written, the target audience appears to be very niche and am not sure if HCI/ Info Viz audiences will be able to see the novel design contributions of this work.

---

### Official Review · Reviewer_PjAo · 2021-05-04
**Prototype AR tool to assist with surgery**

**Rating:** 6
**Confidence:** 3

**Review:**

The paper presents a medium-fidelity prototype of an augmented reality tool to help surgeons navigate during laparoscopic kidney surgery (where the surgeon cannot physically see the area where they are operating).  The prototype adds a visual representation of the kidney and audio feedback indicating proximity to certain features.   The authors conduct a proof-of-concept user study with 11 medical students with a simplified task that only requires targeting indicated areas.  They find that the visual feedback improves accuracy and perceived task workload.  Audio feedback appears to decrease performance when combined with visual feedback.

While preliminary in nature, the paper presents an interesting prototype and the results of the study lead to interesting further questions about the relationship between the visual and audio feedback.   I think the paper is worth publishing, and should generate discussion at the conference.  In particular, the discussion section raises several issues worth exploring.

That said, there are places that could be improved.

The paper is well written, but it is very dense/terse and has a lot of medical terminology.  Some medical terminology is necessary given the topic, but it's not clear that this was written for an HCI audience in mind.  I hardly ever advocate for a paper to be made longer, but in this case, I think that adding some extra explanatory details would be helpful.  Some of the medical terminology could likely also be reduced (or at least defined) to focus of the paper on the HCI aspects of the prototype/idea.

The Results and Discussion sections are unusually divided, with the results being only 3 sentences long (+ 2 tables and figure).  Additional details in the results would help the reader.   Also, were posthoc statistical tests done after the ANOVA to identify which pairs were significant?

Please include additional details about the prototype itself -- why were these designs selected?  Were any HCI principles/theories used to inform the designs?
Please include additional detail about the Control condition, what do participants see in this condition?
Please provide additional detail about the overlap between this paper and the anonymized related paper mentioned on P.5
Was REB clearance received to do the study?
What are the main contributions from an HCI perspective?

---

### Meta-Review · Area_Chair_f2kv · 2021-05-06

**Recommendation:** Accept
**Confidence:** 3

**Metareview:**

All three reviewers are in general agreement about the strengths and weaknesses of the paper.

Strengths:
- the paper presents a prototype of an AR visualization and audio navigation system meant to help during laparoscopic surgical procedures of the kidneys.
- results of the study show that these cues helped to improve targeting accuracy and reduced perceived task workload
- the work would be interesting and relevant to a GI audience, particularly some of the questions/issues raised in the paper's Discussion.

Could be improved:
- the paper uses a lot of medical jargon that would only be understood by a niche audience
- the paper lacks details about the system design process and the user study that would be relevant for a GI/HCI/AR audience
- the authors should clarify the HCI contribution of the work and position the work in the context of existing HCI/AR/Visualization literature
- the results/discussion could be reorganized to include more of the pertinent direct interpretation/explanation of the numbers in the tables/figures directly in the results, and use the Discussion section for the more general, bigger picture items.

On the balance, however, all three reviewers rated the paper positively and agree that it should be published.
We encourage the authors to make some of the suggested revisions to the paper before publication.

---

### Decision · Program_Chairs · 2021-05-08

Accept